# Spatiotemporal Evolution and Driving Mechanism of “Production-Living-Ecology” Functions in China: A Case of Both Sides of Hu Line

**DOI:** 10.3390/ijerph19063488

**Published:** 2022-03-15

**Authors:** Yu Chen, Mengke Zhu

**Affiliations:** 1School of Economics and Management, Zhengzhou University of Light Industry, Science Avenue 136, Zhengzhou 450000, China; 2012030@zzuli.edu.cn; 2School of Public Administration, Huazhong Agricultural University, Wuhan 430070, China

**Keywords:** Hu Line of land, “production-living-ecology” coordination, spatial heterogeneity, ESDA-GWR, China

## Abstract

In order to explore the spatiotemporal evolution of land use function and its driving factors in China, taking both sides of the Hu Line as an example, we used Exploratory Spatial Data Analysis and Geographically Weighted Regression methods to reveal dynamic evolution law, spatial characteristics and influencing factors of the “Production-Living-Ecology” functions of 288 prefecture-level cities on both sides of the Hu Line. The results show that: (1) In the temporal dimension, the coordination of “Production-Living-Ecology” functions of land use in China has been improved, and the Hu Line can be roughly used as the boundary of China’s territorial space use. (2) In the spatial dimension, there is a significant positive spatial correlation between “Production-Living-Ecology” functions of land use in China, and the coordination gap between “Production-Living-Ecology” functions of land use on both sides of the Hu Line is gradually narrowing. (3) In terms of influencing mechanism, the coordination of “Production-Living-Ecology” functions is mainly driven by internal factors and is supplemented by external ones. The influence pattern of most driving factors is consistent with the layout characteristics of “strong east and weak west” of the Hu Line.

## 1. Introduction

In 1935, Hu Huanyong initiated the baseline—“Hu Line” [1]—or revealing the differences in China’s population distribution. In essence, the Hu Line reflects the high spatial coupling between Chinese population and physical geography. It also highlights the problem of uncoordinated regional development in China’s large-scale territorial space. Hu Huanyong believes that differences in natural geographical environment, social and historical conditions, and levels of economic development are the main reasons for the disparity in development between the two sides of the baseline, which coincides with the multi-functionality of land use. Based on diversified regional development goals and development needs at different levels, and under the guidance of natural, social, economic and other conditions, human beings reconstruct the original ecological space on the earth’s surface, so that land use presents multi-functional characteristics such as life, production and ecology [2,3]. However, due to the inherent scarcity of land resources and the equality of opportunities for each function, conflicts between different functional spaces occur frequently, which further aggravates the confusion of the spatial development pattern of the country [4,5]. The main manifestation is that the development intensity of land space with the Hu Line as the boundary has obvious development characteristics of “strength in the east and weak in the west” [6]. As an important ecological function area, the northwest half of the wall has extremely limited space for gathering people and industries. Compared with the southeast half of the wall, the coordination degree of multi-functional land use is lower. Based on this, this paper proposes the following hypothesis:

**Hypothesis** **1.***The functional coordination of the land on both sides of the Hu Line has the characteristics of “high in the east and low in the west”*.

In 2014, Chinese Premier Li Keqiang put forward the issue of “how to crack the Hu Line” [7], reflecting the urgency to clarify the specific factors causing the large gap between the two sides of the Hu Line and eliminate the actual development behind it. As a complete system, land is based on the background of land resources, with human beings as the main body and implementers; it has the characteristics of diversification of internal elements, diversity of utilization processes, and complexity of interaction with society, economy, and ecosystems. The internal subsystems also have synergistic and coherent effects [8]. Therefore, when researching the specific influencing factors of the functional coordination of “Production-Living-Ecology” functions, not only should it be based on the perspective of the whole system, but also the importance of the coordinated development of various subsystems within the system should be considered. Therefore, this paper proposes the following hypothesis:

**Hypothesis** **2.***The functional coordination of land “Production-Living-Ecology” Functions is not only affected by external factors, but also by the coordinated development of various subsystems*.

Based on this, this paper scientifically measures the functional coordination and spatial evolution law of the “Production-Living-Ecology” functions on both sides of the Hu Line, deeply explores the driving factors that cause the difference in coordination, and proposes targeted solutions, which will help alleviate the contradiction between the needs of utilization of the current land, helps to break the “Hu Line” in the national land space. The structure of this paper is as follows: Section 2 is a review of the relevant literature, covering the specific content of the Hu Line and the “Production-Living-Ecology” functions of land use; the third section introduces the research methods and the evaluation indicators of the coordination of the “Production-Living-Ecology” functions on both sides of the Hu Line. The fourth section analyzes the overall characteristics of the coordination of China’s “Production-Living-Ecology” functions and the local characteristics of the southeastern half and the northwestern half bounded by the Hu Line from the perspective of space and time, focusing on the research on the two sides from the internal driving and external driving. Finally, the reason for the spatial heterogeneity of the coordination of “Production-Living-Ecology” functions is drawn, and based on this, countermeasures and suggestions are put forward for breaking through the “Hu Line of Land”.

## 2. Literature Review

### 2.1. Hu Line

At this stage, the research on the Hu line mainly focuses on the extension of its connotation and the exploration of breakthrough paths.

#### 2.1.1. Proposed Hu Line

In 1935, Hu Huanyong first quantitatively described the uneven distribution of China’s population and proposed the famous Aigun Heilongjiang–Tengchong Yunnan population geographic dividing line [9], also known as the “Hu line” [10]. As a geographical boundary of population, the comprehensive geographical basis behind the Hu Line is the suitability and limitation of China’s human settlement environment [11]. Furthermore, China’s ecological carrying capacity has a trend of increasing gradient along the vertical direction of the Hu Line. It can be seen that the Hu Line symbolizes the transformation of China’s ecological environment. As a “virtual boundary”, Hu line is not actually “land”. It is a high-level summary of various geographical boundaries and a deep integration of various geographical mutations, so it is also regarded as a natural geographical boundary in China [12]. In addition, the GDP, which reflects the economic development level of each geographic region, is also highly coupled with the Hu Line. Therefore, the Hu Line is also considered to be the economic geographic boundary of China [13].

#### 2.1.2. Breaking through the Hu Line

The academic circle has come to two main viewpoints around “how to break through the Hu Line”: First, there is the theory that the Hu Line cannot be broken through. From the perspective of natural environment, due to the profound geographical background on both sides of the Hu Line, it is difficult to get rid of the shackles of the environment even if measures such as technological reform and industrial upgrading are implemented. From the perspective of economic development, the difference in transportation costs caused by geographical factors is difficult to eliminate. Therefore, the competitiveness of commodity markets on both sides of the Hu Line is quite different, and this regional economic distribution law is unbreakable [14]. From the perspective of social development, China’s population distribution is highly stubborn, and the gap between the two sides in basic public services, cultural and educational levels will also lead to the long-term existence of the Hu Line. Second, the Hu Line can break through. According to Krugman’s point of view, the second geographic nature is the driving force to overcome the first geographic nature [15], and the development of transportation makes the population and industry agglomerate, making the west open up. In addition, informatization is the third geographic nature, which can promote trade development, knowledge spillovers, and promote preferential development in certain regions [16,17]. The construction of the “One Belt, One Road” has driven the development of the East-West linkage of the Eurasian continent, so that the northwestern half of the country has been transformed from the end of the previous opening to the east to the bridgehead opening to the west.

### 2.2. “Production-Living-Ecology” Function

At present, the research on the “production-living-ecology” function of land use mainly focuses on three aspects: concept definition, classification system and evaluation research.

#### 2.2.1. Origin and Definition of “Production-Living-Ecology” Function

The function of “production-living-ecology” is the extension and development of the multi-functional concept of land use [18]. The multi-functional research on land use originated from the European Union’s agricultural multi-functional research at the end of the 20th century [19], In 1994, the concept of Agricultural Multifunctionality was first used in Uruguay Round Agreement on Agriculture (URAA), and later, OECD defines it as “besides the functionality of food production, agriculture also has the functions of environmental protection, landscape functionality, rural employment, and food safety” [20,21]. Although academia has further expanded agricultural versatility into multi-functional land use [22,23,24], with the formation and promotion of the concept of sustainable development, the concept of land use multi-function emerges at the right moment. Sustainability Impact Assessment: Tools for Environmental Social and Economic Effects of Multifunctional Land Use in European Regions (SENSOR) project defines its concept as “human products and services in the process of land use in a certain region” [25], thus deriving three functional systems of society, economy and environment [26,27]. The “production-living-ecology” function is the product of the coordination and coupling of the three major systems. It further complements the multi-functional characteristics of land use and is regarded as a comprehensive system composed of production, living and ecological functions [28,29].

#### 2.2.2. Classification System of “Production-Living-Ecology” Function

Functional classification of “production-living-ecology” is a process of discovering, representing, naming and classifying land functions [30]. It roughly includes three perspectives: ecosystem function, landscape function and land use [31,32]. Specifically, ecosystem functions derive from the ecological framework [33], but a comprehensive ecosystem service classification system has not been formed [34], and most scholars agree with the 17 ecosystem functions proposed by Costanza based on the summary of ecosystem [35]. Landscape functions evolve on the basis of landscape ecology and spatial planning [36,37]. The classification of landscape functions is very similar to that of ecosystem functions, and de Groot forms the landscape functional classification framework after adding the carrying function according to the original ecosystem functional classification system [38]. In addition, landscape functions are subdivided into three functions: production, ecology and culture [39]. The functional division of the perspective of land use is mainly economic-oriented, which is the product of the combination of land background elements with a certain natural physical and chemical structure and the form of human utilization [40]. According to the mapping relationship between land use structure and function, land use types are classified as “production, living (society) and ecology” [41]. However, this classification method can only reflect the main function of land use and fails to fully show the multi-functional characteristics of each piece of land.

#### 2.2.3. Evaluation Research on “Production-Living-Ecology” Function

The research on the functional evaluation generally includes three parts: analysis framework, evaluation method and influencing factors. From the perspective of analysis framework, the conceptual framework of land use function developed by SENSOR project for analyzing land sustainability incorporates key indicators such as economy, culture and environment at the regional level [42], which greatly develops the multi-functional evaluation method system of land use. Due to the consideration of regional differences, a variety of evaluation frameworks emerge at the historic moment. The integrated analysis framework and participatory framework [43] that integrate various models are gradually becoming a trend. As the basis of evaluation research, the index system mainly covers functional classification and hierarchical design. In terms of specific methods, the comprehensive index method is the most widely used evaluation method. In addition, spatial analysis techniques or mathematical models are usually used to characterize the evolution of land functions [44,45,46]. In terms of influencing factors, the transformation of land use versatility is closely related to natural resource endowment, social and economic conditions and policy factors, but regional policy is the key cause [47]. At the same time, the influencing factors are also dynamic, which is manifested by the factors of environmental resources becoming increasingly prominent [48].

### 2.3. Summarize

To sum up, the Hu line has greatly enriched its connotation, but its role in dividing the spatial pattern of land use has been ignored. Moreover, most of the existing studies have not emphasized the rational layout and long-term development of land use function of land space. In terms of “Production-Living-Ecology” functions, the classification system experienced a trend from simple to complex, from one to multiple, but did not form a “universal” classification system. In terms of evaluation research, the trend of interdisciplinary analysis framework has emerged. Although the selection of indicators is relatively comprehensive, there is a lack of multi-scale research. Most of the influencing factors are qualitative explanations, and lack of analysis of regional differences. Compared with the existing literature, the innovations and contributions of this paper are as follows: (1) At the research scale, the spatial law of the coordination of “Production-Living-Ecology” functions of cities on both sides of the Hu Line is explored from the prefecture-city level, which not only extends the connotation of the Hu Line, but also breaks through the limitations of previous studies on “Production-Living-Ecology” functions focusing on large-scale studies. (2) In terms of research methods, the GWR model was adopted to seek ways to improve and optimize the land use structure on both sides of the Hu Line, which to some extent made up for the lack of discussion on the internal differences of different regions in previous studies. (3) In terms of influencing factors, on the one hand, influencing factors are set on the basis of fully considering the difference between the two sides of the Hu Line. On the other hand, the territorial space system is regarded as a complex dynamic giant system with multiple factors interacting with each other, and internal and external influencing factors are selected comprehensively. (4) Path exploration: Starting from territorial space utilization and starting from “Production-Living-Ecology” functions, it provides a new idea for the study of “how to break through Hu Line”.

## 3. Materials and Methods

### 3.1. Study Area

China has a vast territory, due to the huge difference in resource endowment between the east and the west, resulting in a “strong east and weak west” distribution pattern in population, economy and other aspects with the Hu Line as the boundary. As shown in Figure 1, the population and economy west of the Hu Line only accounted for 7% of the southeast half in 2010. Although the proportion rose to 7.3% and 7.6%, respectively, in 2020, this pattern still had no substantial change in 2020. In recent years, with the continuous promotion of the main function zoning and new urbanization development strategy, the pattern of land use production, living and ecological function between urban and rural areas has changed dramatically, the strengths and weaknesses of the “Production-Living-Ecology” functions and the interlocking situations also have obvious regional colors, showing different development states on both sides of the Hu Line. There are few studies in the literature on the Hu Line in delineating the coordination characteristics of the “Production-Living-Ecology” functions of land use. Based on the availability of data, this paper selects a total of 288 prefecture-level cities on both sides of the Hu line as the study sample, involving a total of 30 provinces and autonomous regions except Tibet and three municipalities directly under the central government, Beijing, Shanghai and Tianjin, which account for 88% of China’s total population and 91% of China’s GDP; therefore, the results obtained using these cities have a wide range of practicality and play a role in supporting the scientific exploration of the theory of the spatial layout of China’s land from the perspective of Hu line.

### 3.2. Research Method

#### 3.2.1. Coupling Coordination Analysis

Coupling coordination degree can comprehensively reflect the overall function of environment and economy or the coordinated development level of comprehensive environment and economy. Referring to the existing research [49,50], this paper constructs the coupling and coordination model of production, living and ecological functions of land use to quantitatively identify the coordinated development level of “production-living-ecology” function and realize multi-scale integrated evaluation and analysis. See Equations (1)–(3) for calculation method:(1)$$C={\left\{\frac{f_{1}\times f_{2}\times f_{3}}{\left(f_{1}+f_{2})\times (f_{1}+f_{3})\times (f_{2}+f_{3}\right)}\right\}}^{1/3}$$
(2)$$T=\alpha f_{1}+\beta f_{2}+\gamma f_{3}$$
(3)$$D=\sqrt[{}]{C\times T}$$

In the formula, *C* is the coupling degree; *f_i_* is the comprehensive evaluation function of each function; *T* is the comprehensive evaluation value of “production-living-ecology” function, reflecting the overall coordination effect; *α*, *β*, *γ* are undetermined weights. In this paper, since the “production-living-ecology” function of land is equally important, *α* = *β* = *γ* = 1/3; *D* is the coupling coordination degree, the greater the *D* value, the better the coupling coordination. Among them, coupling degree refers to the degree of mutual influence of land “production-living-ecology” functions; coupling coordination degree reflects the level of harmony between the “production-living-ecology” functions in the land use function system in the development process.

#### 3.2.2. Spatial Pattern Analysis

Exploratory spatial data analysis (ESDA) is an ideal data-driven analysis method. The essence of the model is to use a series of spatial data analysis methods and technologies, with spatial relevance as the core, through the description and visualization of the spatial phenomena, to find spatial agglomeration and spatial anomalies, thus revealing the spatial interaction mechanism between research objects. ESDA is divided into global spatial autocorrelation and local spatial autocorrelation. The global Moran’s I index is used to describe the overall spatial characteristics of efficiency, so as to judge the spatial correlation and difference characteristics; the local *Gi** index is used to describe the local spatial heterogeneity characteristics of efficiency, so as to judge the local spatial differentiation laws. In this paper, global Moran’s I and local *Gi** indexes are used to measure the spatial pattern characteristics of “production-living-ecology” function coordination degree at the city level in China.

##### Global Spatial Autocorrelation

The global Moran’s *I* index measures the general trend of spatial correlation of the unit attribute values of adjacent or similar regions in the whole study area.
(4)$$I=\frac{\sum_{i=1}^{n}\sum_{j=1}^{n}W_{ij}(X_{i}-{\bar{X}}_{})(X_{j}-{\bar{X}}_{})}{S^{2}\sum_{i=1}^{n}\sum_{j=1}^{n}W_{ij}}$$
where: *n* is the number of research objects; *X_i_* and *X_j_* represent the observation values of *i* and *j* regions, respectively; *W_ij_* is the spatial weight matrix (1 for spatial adjacency and 0 for non-adjacency); *S*^2^ is the variance of observation values; and is the average of observation values. At a given significance level, a positive Moran’s *I* value means that the overall coordination degree of urban “production-living-ecology” function coupling shows significant spatial agglomeration characteristics; if Moran’s *I* value is negative, it means that the overall coordination degree of urban “production-living-ecology” function coupling shows significant spatial differentiation.

##### Hot Spot Analysis

It is used to analyze the hot and cold regions in different spatial regions, so as to measure the autocorrelation characteristics of local space.
(5)$$G_{i}^{*}=\sum_{j=1}^{n}W_{ij}\frac{X_{i}}{\sum_{j=1}^{n}X_{j}}$$
where: *W_ij_* is the spatial weight matrix, the spatial adjacency is 1, and the non-adjacency is 0. If it is significantly positive, it indicates that the value around *i* is relatively high, which belongs to the hot spot area; otherwise, the value around *i* is relatively low, which belongs to the cold spot area.

##### Driving Factor Analysis

Compared with the traditional OLS regression model, geographic weighted regression (GWR) uses spatial relations to reflect the non-stationary characteristics of parameters in different spatial locations, so that the relationship between research variables changes with the change of spatial location. Therefore, it can be used to reflect the spatial heterogeneity of the impact of different factors on the coordination of “production-living-ecology” function of land use. The model structure is as follows:(6)$$y_{i}=\beta_{0}(u_{i},v_{i})+\sum_{k}\beta_{k}(u_{i},v_{i})x_{ik}+\varepsilon_{i}$$

In Equation (6), *y_i_* is the dependent variable value at the geographic location (*u_i_*, *v_i_*), (*u_i_*, *v*_i_) is the geocentric coordinate of the sample space unit, *β*_0_(*u_i_*, *v_i_*) is the constant value at the geographic location (*u_i_*, *v_i_*), *β*_k_(*u_i_*, *v_i_*) is the value of function *β*_k_(*u*, *v*) at the location of *I*, *ε_i_* is the spatial random residual.

### 3.3. Index System Construction and Data Sources

The huge difference in natural geographical environment on both sides of the Hu Line and the inherent multi-dimensional, complex and scarce attributes of land space directly lead to the diversity of land functions on both sides. The 18th National Congress of the Communist Party of China proposed that the development and utilization of land space should be in line with “intensive and efficient production space, moderate livable space and picturesque ecological space”. Therefore, on the basis of integrating the huge differences in production, life and ecology on both sides of the Hu Line, this paper constructed a three-dimensional index system (Table 1) of “production-living-ecology” from top to bottom according to the “system-element-function” theory and the “production-living-ecology” spatial optimal group theory [51]. First of all, the spatial distribution of population represented by the Hu Line is highly consistent with the spatial distribution of productive land resources, and the basic goal of production function is to achieve spatial concentration of production and guarantee social and economic development. Based on the mode of production and the level of economic development, agricultural production and non-agricultural production represent the mode of production. The former relies on agricultural land to obtain material products, while the latter relies on construction land to obtain commodities and services, therefore, the corresponding indexes are selected. From the perspective of the internal and external environment of economic development, indicators such as per capita GDP and amount of foreign capital used are selected to represent. Secondly, the Hu Line is also the demarcation line of human livability, and the basic goal of life function is to promote the concentration of living space and improve the livability level of cities. The improvement of living function cannot be separated from the people-oriented development concept. Based on Maslow’s hierarchy of needs theory, the living standard, material life and spiritual life respectively represent the physiological, safety and self-actualization needs that people need to meet for survival. Living standards are reflected in the security of life, such as the need to ensure housing and travel. Material life is embodied in material wealth, such as the level of employment and consumption. Spiritual life is reflected in the provision of spiritual services, such as scientific and educational investment and cultural service provision. Therefore, the corresponding indicators are selected. Finally, the Hu Line symbolizes the transformation of China’s ecological environment, and the basic goal of ecological function is to promote the integration of ecological space and maintain human living conditions. Ecological function is the prerequisite for the realization of production and living function and the natural base of land use [52]. The corresponding indicators are selected based on ecological foundation, carrying capacity and governance capacity. Based on the above analysis, this paper constructs an evaluation system of 27 basic indicators with 9 criteria levels.

## 4. Results

### 4.1. Analysis on the Coordinated Space-Time Pattern of “Production-Living-Ecology” Function from the Perspective of Hu Line

This section includes two parts: characteristics of spatio-temporal changes of “production-living-ecology” function coordination and spatial differentiation of driving factors.

#### 4.1.1. Characteristics of Spatio-Temporal Changes of “Production-Living-Ecology” Functional Coordination

According to Equations (1)–(3), calculate the coordination degree of “Production-Living-Ecology” function of 288 prefecture-level cities on both sides of the Hu Line from 2008 to 2017 (Table 2), and analyze the time variation trend of coordination degree of “Production-Living-Ecology” function on both sides. Overall, the coordination of “Production-Living-Ecology” function in China has improved as a whole, rising from 0.1207 in 2008 to 0.1364 in 2017, an increase of 13% in 10 years, but the growth rate is relatively slow. From the point of view of the Hu line, the temporal variation characteristics of the coordination of “Production-Living-Ecology” function on both sides are basically synchronous. According to the data, the coordination of “Production-Living-Ecology” function on both sides of the Hu Line is characterized by “high in the east and low in the west”. Although the coordination difference between the two sides fluctuated, decreasing from 0.0154 in 2008 to 0.026 in 2017, in general, the coordination of “Production-Living-Ecology” function in the southeast half of the wall has a certain advantage compared with the northwest half. This is consistent with the basic situation of regional economic development. As the southeast half of China, which is economically developed, the first round of industrialization oriented by market economy has come to an end, while the new round of industrialization based on the principle of “industrial agglomeration, centralized layout and intensive land use” is booming. In contrast, the northwest half of China is in a transitional stage from the early stage of industrialization to the middle stage and needs to rely on a large amount of land input to promote the rapid development of industrialization, resulting in a relatively weak coordination of land use functions.

ArcGIS10.0 software (Enveronmental Systems Research Institute (ESRI), RedLands, CA, USA) was used to generate the spatial distribution pattern map (Figure 2) to analyze the spatial changes of the coordination of Production-living-Ecology function of land use in China. The results are as follows:(1)Low-level coordination area. In 2017, this type of area accounted for 44.8%, an increase of 10.8% compared with 2008, showing a significant change. Most of them appear in the west of the Hu Line and the three major forest areas, mostly in a continuous situation. Except for a few provincial capitals and municipalities, they are all at a low level. Among them, the western region accounted for 60.2% in 2008 and decreased to 48.1% in 2017. In contrast, in the east, a large number of cities have seen a significant deterioration in the functional coordination of “production-living-ecology”, and the proportion of low-level coordination areas has increased from 11.2% to 23.2%. The central region has not changed much.(2)Primary coordination area. From 2008 to 2017, there was no significant change in the primary coordination area, and the proportion declined slightly. Among them, the proportion in 2008 and 2017 was 38.5% and 35.8%, respectively. They were distributed on both sides of the Hu Line and mainly concentrated in the central and eastern regions of China, accounting for about 40%. In these areas, the “core-periphery” change pattern mostly appears, that is, the trend of decreasing outward around the advanced or intermediate coordination area, and the most obvious is in the middle and lower reaches of the Yangtze River and the Bohai Rim region.(3)Intermediate coordination area. Contrary to the low-level coordination areas, the intermediate coordination areas are mainly distributed east of the Hu Line, and the proportion of cities in the intermediate coordination areas decreased significantly from 2008 to 2017, from 65% to 43%, Although the change in the proportion of each area is not significant, the cities included have decreased to varying degrees, especially in the central and eastern regions, which decreased from 21 and 35 in 2008 to 13 and 23 in 2017, respectively. Among them, the Loess Plateau and the middle and lower reaches of the Yangtze River have the most significant changes from agglomerates to sporadic distribution.(4)Advanced coordination area. From 2008 to 2017, the proportion of advanced coordination has not changed significantly, all of which are below 5%. East of the Hu Line is the most important distribution area, and most of these cities are located within the urban agglomeration. Most regions also occupy the position of central cities in the urban agglomeration. For example, Beijing, Shanghai, and Zhengzhou belong to the core cities of Beijing-Tianjin-Hebei, the Yangtze River Delta, and the Central Plains urban agglomeration. In addition, compared with 2008, the advanced coordination area has changed from a simple division in the central and eastern regions to the western and middle eastern regions, and the advanced coordination area in the west is also located within the urban agglomeration on the northern slope of Tianshan Mountain.

The above analysis shows that: First of all, the Hu Line can be interpreted to some extent as the boundary of China’s territorial space utilization. With this line as the boundary, the west is mostly the low-level coordination area of land use “production-living-ecology” function, while the east is mostly the intermediate and advanced coordination area. The main reason is that the Hu Line can be used as the boundary of China’s ecological environment, and the northwest is the main ecological spatial distribution area in China. Industrialization and urbanization, as the basic driving force for the transformation of “production-living-ecology” functions, are subject to their special landforms, resulting in a large gap among production, living and ecological functions. However, the southeast has significant advantages in public services and industrial development, and its land use is more reasonable due to the increasingly intensified ecological protection policies in recent years. Secondly, from the perspective of the three major regions in China, the proportion of the low-level coordination areas in the west has decreased significantly. The main reason may be that the implementation of the western development policy on the one hand promoted regional economic growth and paid more attention to ecological restoration and environmental protection, thus promoting the gradual rise of production and living functions. The proportion of intermediate coordination areas in the central and eastern regions decreased significantly. This is mainly because with the development of industry, some low-end industries have infiltrated into and threatened the development of ecological environment. Although policies such as “ecological civilization construction” and “main function zoning” have effectively restricted the development path at the expense of the environment, they have also led to the continuous compression of production and life functions. The advanced coordination area is mostly located in the central city of the eastern urban agglomeration and shows the attenuation trend that the advanced coordination area as the center gradually weakens to the outer layer. By forming a central city with strong radiation power, urban agglomerations promote the development of surrounding and peripheral cities, and form an urban agglomeration system with reasonable layout, division of labor and cooperation, and complementary functions. It is increasingly becoming the main spatial form of various development factors and the key area for industrialization and urbanization development. With many advantages, its function coordination of “production-living-ecology” can achieve rapid development.

Therefore, the above analysis can verify hypothesis 1: the coordination of land Production-Living-Ecology function on both sides of the Hu Line is characterized by “high in the east and low in the west”.

#### 4.1.2. Evolution of Spatial Pattern of “Production-Living-Ecology” Functional Coordination

In order to quantitatively study the evolution of the spatial pattern of Production-Living-Ecology function coordination, this paper established Queen’s adjacency matrix and used the data of prefecture-level cities from 2008 to 2017 to calculate the Global Moran’s I of the study area over the years. It was found that the Moran’s I index passed the test at the significance level of 0.01 in all years, indicating that the coordination level of Production-Living-Ecology function of prefecture-level cities in China had a positive spatial autocorrelation. In addition, the global Moran’s I indices in 2008 and 2017 were 0.1287 and 0.1034, respectively, indicating that the spatial distribution pattern of the coordination level of Production-Living-Ecology function in China was relatively stable in general.

To eliminate global autocorrelation to local instability of defects, and further detect the local spatial agglomeration pattern evolution characteristics, ArcGIS10.0 software (Enveronmental Systems Research Institute (ESRI), RedLands, CA, USA) was used to calculate the 2008 and 2017 in the local *Gi** index of each city. The natural breakpoint method is used to divide the values into hot spot area, second hot spot area, second cold spot area and cold spot area, and draw the clustering and evolution chart of the “production-living-ecology” functional coordination degree pattern of cities in China from 2008 to 2017 (Figure 3).

On the whole, there has been no substantial change in the pattern of cold and hot spots in the coordination degree of “production-living-ecology” in China’s cities, and the situation of “high in east and low in west” is still maintained on both sides of the boundary of the Hu Line. Among them, hot spots did not form an obvious global high-value concentration center. Most of the hot spots in 2008 were located on the east side of the Hu Line (Qiqihar, Shenyang, Dalian, Beijing, Cangzhou, Zhengzhou, Wuhan, Nanjing, Wuxi, Suzhou, Shanghai, Xiamen, Guangzhou and Shenzhen). Most of them are provincial capitals. In 2017, although cities in the southeast (Benxi, Cangzhou, Zhengzhou, Wuhan, Nanjing, Shanghai, Guangzhou, Dongguan and Shenzhen) are still dominant, they moved westward and spread to Urumqi and Karamay on the west side of Hu Line, and some regions were reduced from hot spots to second hot spots or cold spots. At the same time, there is also a certain degree of spatial polarization in some regions, such as the Central Plains urban agglomeration and the Yangtze River Delta city group, whose agglomeration trend shows an attenuation trend from east to west. The cold spot area gradually moves to the east and spreads, and presents an obvious global agglomeration state, especially in areas such as the Loess Plateau and Sichuan Basin, accompanied by the transformation of many second cold spot areas into cold spot areas. From the overall change of the coordination degree pattern of “production-living-ecology” function from 2008 to 2017, the differences between the two sides of the Hu Line are still significant. An overall high-value concentrated central area has not been formed, which indicates that there is a lack of “leading” cities in China to promote the regional coordination of “production-living-ecology”. In addition, the expansion of cold spots and second cold spots, as well as the compression of hot spots and second hot spots, indicating that the “production-living-ecology” functional coordination gap on both sides of the Hu Line in territorial space planning is gradually narrowing.

### 4.2. Research on Spatial Differentiation of Driving Factors

On the basis of determining the driving factors, an analysis of the spatial heterogeneity of driving factors is carried out.

#### 4.2.1. Driving Factor Analysis Framework

As a complex dynamic giant system interacting with multiple elements [53], the land space system is not only strongly restricted by the relationship between subsystems within the system, but also constrained by many natural, social and economic elements outside the system. Therefore, this paper starts from the perspective of system theory, especially selects the production-living function, production-ecology function and living-ecology function to study the influence of each subsystem (Table 3). At the same time, considering the differences of human and natural environments on both sides of the Hu Line, population density, financial density, economic density and water resource density are selected as external environmental factors to study their influence on the coordination of “production-living-ecology” functions. From the perspective of an internal driving force, the coordination between two subsystems will directly affect the development of the overall coordination of “production-living-ecology” function. Among them, the coordination of production-living function means that the stronger production function is not only the economic development, but also meets the needs of infrastructure construction and social security improvement, thus strengthening the living function. The coordination of production-ecological function means that the development of industrial parks and new technologies enhances the intensity of industrial agglomeration and provides greater development space for ecological function. The coordination of living-ecological function means that the living function and ecological function can be improved synchronously by relying on powerful resources such as culture and infrastructure. From the perspective of external driving forces, the Hu Line has long been known as the dividing line of China’s demographic geography, economic geography and physical geography due to huge differences in population distribution, economic development and natural environment. First of all, as the boundary of population geography, the Hu Line divides China into densely populated areas and sparse areas, so population density is chosen to represent the population driving effect. The increase in population will directly aggravate the scarcity resources in the system, and then cause obvious effect on the change of production, living and ecological function structure. Secondly, China’s fiscal expenditure layout is coupled to the Hu Line to a certain extent, so the financial density is chosen to represent the investment-driven effect. As a factor of production, fiscal density directly enters into the economic production sector to improve the output level of various industries, as a financial support for developing infrastructure and ensuring people’s livelihood, and indirectly increases output by improving ecological environment and production conditions. Thirdly, as the economic geographical dividing line, the economic development level of both sides of the Hu Line is always high in the east and low in the west. Therefore, economic density is chosen to represent economic driving effect. Areas with a high level of economic development tend to have a high level of intensive land use, which will promote the coordinated development of the “production-living-ecology” function. The area with a low level of economic development may also rely on “backwardness advantage” to obtain tremendous development. Finally, the Hu Line, as the boundary of abrupt ecological changes, divides China’s semi-humid region and semi-arid region, and basically coincides with the 400mm annual rainfall line. Therefore, water resource density is chosen to represent water driving action. This factor restricts the development of “production-living-ecology” function through ecological water, production water and domestic water, and makes the three interact and combine organically with different water resource benefits. Therefore, this paper analyzes the driving mechanism of the “production-living-ecology” function of the national space system through two driving factors, internal and external.

#### 4.2.2. Spatial Heterogeneity of Driving Factors

Unlike ordinary linear regression models, GWR considers the influence of different spatial positions on the regression results, which can be used to explore the non-stationarity of spatial relations. Geographically weighted regression model was used to set ADAPTIVE and AICc as bandwidth, and R2 was 0.7939, indicating that the model had a good overall fitting effect and could well simulate the influence of variables on the coordination of “Production-Living-Ecology” functions. The results show that each variable has different estimation results for different cities, indicating that there are spatial differences in the impact on the coordination of “Production-Living-Ecology” Functions in different regions (Table 4). This kind of influence is mainly driven by internal, and auxiliary by external. This also verifies hypothesis 2 of this paper, that the coordination of land “Production-Living-Ecology” Functions is not only affected by external factors, but also by the coordinated development of subsystems, as follows:

Production-Living function (Figure 4a) coordination has a two-way effect on “Production-Living-Ecology” Functions coordination, but it is mainly positive, 98.6% of prefecture-level cities have a positive correlation with “Production-Living-Ecology” Functions coordination. There is a certain spatial coupling with the Hu Line. The mean value of the regression coefficient of the southeast half wall is 0.1864, which is significantly higher than that of the northwest half wall, and most of the high value areas are located in the areas east of the Hu Line, with a bipolar distribution, respectively, in the northeast and some southern areas. In recent years, under the guidance of the revitalization of the Northeast policy, three northeastern provinces region have continued to optimize the production space by promoting the modernization of agriculture and industry, providing the necessary support to improve the functions of the social system and increasing the coordination of Production-life functions. In contrast, the southern high-value areas are mostly poverty-stricken zones. With the support of industrial poverty alleviation policies, a large number of abandoned home bases after relocation to alleviate poverty have been reclaimed, transforming the original construction land into arable land and land for infrastructure construction, and increasing the diversity and coordination of land use functions.

The degree of coordination of production-ecological functions (Figure 4b) also has a two-way influence on the coordination of the “three living” functions, but it is also mainly positive, in which 86.8% of the prefecture-level cities are positively correlated with the coordination of “Production-Living-Ecology” Functions, the spatial distribution of “Production-Living-Ecology” Functions is stepped from the central high value area to the northeast and southwest. The mean values of the regression coefficients on both sides of the Hu Line are 0.0711 and 0.0628, respectively, and most of the high value areas are located on the east side of the Hu Line, especially concentrated in some cities in the central and western parts and the eastern coast. Among them, thanks to the support of the western development strategy and ecological protection project, the ecological construction and economic development in the central and western regions can be synchronized, which is conducive to improving the coordination degree of land use function. In addition, the low concentration in northeastern China, basic negative regression coefficient, while the strategy of rejuvenating northeast to some extent inhibited the economic downturn, but as the old industrial base, its ills, mainly manifested in and out of the land quantity reduction and poor resources and environment development, etc., and the economic growth of locking problems such as path dependence or path is very prominent [54].

The coordination degree of Life-Ecology functions (Figure 4c) has a significant positive impact on the coordination degree of “Production-Living-Ecology” Functions, presenting a decreasing spatial distribution of lumpiness from west to east. Different from the distribution of “high in the east and low in the west” on both sides of the Hu Line, the mean regression coefficient of the southeast half is 0.3924, significantly lower than the 0.5133 in the northwest half. The high value area is also mainly distributed in the west of the Hu Line, among which the cities in Xinjiang are the most prominent. The western development strategy, and beautiful rural construction allows to give priority to the western region of Xinjiang improving infrastructure, combined with recent years dedicated to incorporating the ecological concept in the urbanization construction in Xinjiang, always in a good ecological resources capitalization of opportunity at the same time, the rural residential environment renovation of full swing action, to push the synchronizing lifting of life and ecological function. Low in the area mainly gathered on the Hu Line east of northeast China, the reason may lie in the traditional sense of the urbanization development of negative effects on the ecosystem is easy, but at this stage of construction land continue to increase, unreasonable land use planning problems such as the northeast to continue in a state of ecological deficit, so hindered the coordinated development of “Production-Living-Ecology” Functions.

Economy density (Figure 5a) also has a two-way impact on the “Production-Living-Ecology” Functions, but the regression coefficients of all cities except Xinjiang are positive. In space, it is similar to the pattern of “strong east and weak west” on both sides of the Hu Line, and the mean value of regression coefficient in the east is larger than that in the west. However, the high value area shows an obvious polarization trend, not only concentrated in the Pearl River Delta east of the Hu Line, but also concentrated in the northwest west of the Hu Line. Among them, as the second largest economic zone in China, the Pearl River Delta has become more mature in its economic development. Compared with its response to urbanization, land use change responds more strongly to regional timeliness policies and pays more attention to the coordinated development of land use functions. Unlike the Pearl River Delta, after China’s economy enters the new normal, the rapidly developing urbanization in western China has brought about a continuous increase in the demand for construction land. However, all western regions represented by Gansu province have greatly alleviated the contradiction in land demand at this stage by vigorously developing ecological tourism industry and modern agriculture.

Finance density (Figure 5b) has a significant positive impact on the coordination of “Production-Living-Ecology” Functions, and gradually decreases from the center of the low-value area to the periphery. Unlike the distribution of the Hu Line, the distribution of regression coefficient of fiscal density shows the characteristics of “strong in the west and weak in the east”, and the high value area is distributed on both sides, mainly concentrated in western China and some parts of northeast China, among which the western region dominated by Xinjiang is the most prominent. As a typical area of government-led urbanization, this top-down development mode enables Xinjiang to obtain a great financial tilt, which makes Xinjiang “build a city first, then establish a city” under the government planning, and gradually rationalize the functional structure of land use. On the contrary, fiscal density plays a small role in promoting the coordination of “production-production-production” functions in the central and eastern cities east of the Hu Line, which may be due to the inefficiency and corruption of government financial expenditure in these areas, or the excessively high level of expenditure distorts the function of market resource allocation [55], resulting in low land use efficiency, and further affecting the development and coordination of various functions.

Population density (Figure 5c) also has a positive impact on the coordination of “Production-Living-Ecology” Functions, presenting a stepwise spatial distribution from southeast to northwest. The mean values of regression coefficients on both sides of The Hu Line are 0.0331 and 0.0474, respectively, showing the distribution characteristics of “strong in the west and weak in the east”. The high values are distributed in the northwest region west of the Hu Line and the northeast region east of the Hu Line. Among them, the new population policy and the northeast revitalization plan greatly change the stagnation of population development in the northeast, and direct effects on regional population growth and the migration process, and thus indirectly affect the transformation of land use function, in essence, it is a continuous reconstruction process from conflict to coordination of different land use functions or types in space [56]. The low value area is located in the Pearl River Delta and its surrounding areas on the east side of the Hu Line, where the population and land use show a positive allometric relationship [57], rapid economic growth continues to strengthen the “siphon effect” of population. In order to improve the living standard of urban residents, a large number of high-class residential buildings have been built in many areas, which not only failed to improve the living standard of residents, but also aggravated the disorder of urban sprawl, thus damaging the coordinated development of “Production-Living-Ecology” Functions.

Water resource density (Figure 5d) has a positive and negative bidirectional influence on the coordination of “Production-Living-Ecology” Functions. The east of the Hu Line is mostly positive, while the west of the Hu Line is mainly negative. Xinjiang region is the high value center of negative impact. The special geographical location determines that most of the cities in Xinjiang are typical oasis cities in arid areas. Urumqi and Karamay are the economic development centers at the northern foot of Xinjiang. The rapid development of industrialization leads to the continuous displacement of agricultural water by production and living water, which restricts the development of cultivated land, and the continuous encroachment of production and construction land on the basic ecological land at the outer edge of the city, resulting in the imbalance of spatial distribution of urban land use functions. The positive impact areas are almost all located in the east of the Hu Line, mainly in the Yangtze River, with abundant water resources compared with Xinjiang and other places. Especially, in recent years, the development concept of “joint efforts to protect and not to develop” has not only reversed the situation of ecological environment deterioration in the Yangtze River Basin, but also accelerated the change of industrial structure and strengthened the coordination of regional land multi-functional utilization.

## 5. Conclusions and Discussion

### 5.1. Conclusions

Based on the discussion on the coordination of Production-Living-Ecology function of land use in 288 prefecture-level cities on both sides of the Hu Line from 2008 to 2017, this study mainly verified the two hypotheses proposed in this paper.

Firstly, from the perspective of time dimension, it can be seen that the function coordination degree of “Production-living-Ecology” on the east side of the Hu Line is higher than that on the west side. From the spatial dimension, it is found that the west of the Hu Line is mostly a low-level coordination area, while the east is mostly a middle and high-level coordination area, which verifies hypothesis 1: the coordination of land Production-Living-Ecological function on both sides of the Hu Line is characterized by “high in the east and low in the west”.

Secondly, GWR was used to analyze the driving mechanism that affected the coordination of Production-Living-Ecology function of land use. It was found that there were significant regional differences in the impact of each driving factor on the coordination of Production-Living-Ecology function. The order of influence degree is Life-Ecology function > Production-Living function > Production-Ecology function > Economic density > Finance density > Population density > Water resource density. It shows that the coordination of Production-Ecology function is mainly driven by internal factors and is supplemented by external factors. That is, hypothesis 2 of this paper is verified: the coordination of land Production-Living-Ecology function is not only affected by external factors, but also by the coordinated development of subsystems.

### 5.2. Discussion

According to the research conclusion of this paper, there are three major characteristics of the coordination of the “Production-Living-Ecology” Functions of China’s land use: non-equilibrium on both sides of the Hu line, mainly internally driven and externally driven. Therefore, in order to balance the coordinated use of China’s land space, countermeasures should be suggested from these three characteristics.

#### 5.2.1. Firmly Grasp the Main Line of Coordinated Regional Development and Accelerate the Breaking of the “Land Hu Line”

The concept of “main functional area” was first proposed in 2004 and then incorporated into the regional coordinated development strategy [58]. Therefore, we can learn from its core idea and plan the utilization of the national land on both sides of the Hu Line through the spatial governance model of the main functional area and ensure the coordination and sustainable development of “Production-Life-Ecology” Functions through three levels of action mechanism of “planning, control and evaluation”.

First, delineate the main functional area to ensure the balance of “Production-Living-Ecology” Functions. Through the above research, it can be found that most of the areas west of the Hu Line, mainly Xinjiang and Tibet, are ecologically fragile areas, and the contradiction between economic and social development and ecological protection is more prominent. Therefore, the balanced development of different functions can be achieved by delimiting main functional areas. The land space west of the Hu Line is divided into spatial units with cities and counties as units, and each unit is given a unique main function, thus forming a “one blueprint” for the main functional zoning at the city and county level nationwide; delineate three red lines of urban development boundary, permanent basic farmland and ecological protection, and effectively restrict the production, living and ecological protection behavior of each space by clarifying the spatial boundary of the main functional area. Second, refine the management and control development, and realize the coordinated development of “Production-Living-Ecology” Functions from the process. In the development, the development direction of each area should be clearly defined. In the key development urbanization areas, due to their strong resource and environmental carrying capacity, it is necessary to strengthen the construction of urban functions, guide the orderly clustering of industries, and strengthen the production and living functions of land use; for the main agricultural production areas with restricted development, this type of area is mainly used for the supply of agricultural products, so we should focus on the agricultural production function of land use, and appropriately strengthen its living and ecological functions; for key ecological function areas that are restricted from development, this type of area mainly provides ecological products to ensure national ecological security, and should focus on grasping the ecological function of land use. Third, the assessment of the development performance of each, from the system to ensure that the concept of “Production-Living-Ecology” accurate implementation. The supporting policies of functional areas can be implemented in accordance with the positioning of the main functional areas and categorized and managed. To provide policy support for investment, industry and population in areas with large urbanization needs, especially for industrial development west of the “Hu Line”; for the main agricultural production areas east of the Hu line can improve the living standards of farmers through the agricultural price compensation mechanism, while increasing the investment and construction of industries and public services; for the key ecological function areas west of the Hu line, on the one hand, financial transfer payments should be increased, and on the other hand, industrial access thresholds should be strictly enforced as a way to strengthen ecological functions and appropriately improve production and living functions.

#### 5.2.2. Enhance the Internal Driving Force Coordination and Promote the Coordinated Development of “Production-Living-Ecology” Function

We should further promote the coordination degree of production, living and ecological functions, strengthen the integrated layout of “Production-Living-Ecology” Space, realize the symbiosis and co-prosperity of “Production-Living-Ecology” Function, and ensure the coordinated development of land use functions.

First, we will adopt classified policies to ease the contradiction between the use of land for production and living purposes [59]. Distinguish different industrial types and adopt corresponding layout patterns. For the first-class industrial land, the Production-Life compound layout mode is adopted, while for the second- and third-class industrial land, the independent production space mode can be adopted. The compactness of production space should be strengthened to make production space intensive and efficient based on the layout of industrial chain relations. According to the needs of different groups, improve the layout of infrastructure, strengthen the connection between production and living space, realize the co-construction and sharing of infrastructure, and achieve the overall coordination of production and living functions. Second, upgrade and restructure to promote synchronous development of production and ecological functions. For the Production-Ecology function, the key lies in the upgrading of industrial structure. By increasing industrial investment, enterprises should be guided to develop circular economies to achieve sustainable development, so as to make up for the deficiency of natural ecology with high-quality production. Reconstruct urban structure, strictly delimit land use boundary, maintain cultivated land area by urban growth boundary and cultivated land protection boundary, and restrict the disorderly expansion of urban construction land. To protect green space to isolate industrial areas, so as to reduce urban pollution and improve ecological quality, and ensure that the ecological function of land use can be brought into play. Third, we need to take multiple measures to promote the coordinated development of living and ecological functions. The living environment should be improved, the population size and distribution should be considered comprehensively, and the community life circle should be formed by improving the construction and coverage of public service facilities. Vigorously develop livable cities, build a network ecological landscape pattern of “corridors around the city, green wedge introduction, park Mosaic, multi-corridor connection”, form a complete and homogeneous ecosystem, improve public service facilities and infrastructure on this basis, to create a green livable city. Deployment of pilot city double repair work actively, carry out urban repair and ecological restoration, from is closely related to the residents of “environment” of improving infrastructure, increase investment in environmental governance at the same time, the ecological red line within the scope of the houses or buildings moved or renovation, to carry out ecological restoration systemic reshape city temperament.

#### 5.2.3. Optimize the Influence of External Drivers to Facilitate the Continuous Coordination of “Production-Living-Ecology” Function

On the basis of considering regional heterogeneity, the external impact factors should be optimized according to local conditions to help different regions realize the sustainable development of land use “Production-Living-Ecology” functions.

First, differentiated development strategies should be adopted according to different stages of economic development. The response of land use change to urbanization and economic development level is no longer obvious in some areas east of the Hu Line where economic development has become mature. Therefore, it is necessary to constantly strengthen the government’s function of policy guidance and control of the bottom line, and strictly delimit the ecological red line to give consideration to economic development and ecological protection [35]. For the region west of Hu Line, which is in the stage of economic transformation and development, special attention should be paid to the coupling development of urbanization and industrialization [50], the intensity of development should be controlled in strict accordance with the red line for ecological protection, and a negative list of industrial access should be drawn up to prevent “pollution shelter effect” in key ecological function zones, so as to ensure the ecological function of land use. Second, reform the existing fiscal system to facilitate spatially balanced development. It is necessary to gradually establish a financial system that matches the balanced development of space to support the coordinated symbiosis of various functions of land use. A three-dimensional model of spatially balanced development can be constructed, which roughly includes three levels: the production layer, the secondary distribution layer and the actual consumption layer of residents, in order to meet the inevitable requirements of economic development, we adhere to an unbalanced development strategy at the production layer, and at the same time, we must establish a strong redistribution system at the secondary distribution layer, but this requires the support of a scientific transfer payment mechanism. In addition, it must have a sound legal system and establish a comprehensive spatial planning system to ensure its operation. Third, do a good job in human resource development and reserve, and start the “open source and reduce expenditure” of urban land. For the economically developed areas east of the Hu Line with a large population, we should consciously formulate policies to control population, “open source” on the basis of “reducing expenditure”, strictly control the expansion of urban construction land, and avoid the deterioration of man-land relationship. While the population control policies in some eastern megacities will be effective, they will be accompanied by the local migration of the population in the central and western regions. Therefore, cities to the west of the Hu Line should do a good job in the reserve, development and utilization of various human resources. Fourthly, rational use of water resources to improve the ecological environment. For the areas west of the Hu Line, where water resources are scarce, water resources should be allocated and managed based on the overall interests of the whole basin in accordance with the general idea of “reducing agricultural water, saving domestic water, increasing ecological water and ensuring industrial water”. For water source region of the east of the relatively rich Hu Line, first priority is to insist on protection priority, constantly optimize the ecological environment, in returning farmland to forest and grass measures such as to give full play to the ecosystem on the basis of self-healing, adherence to the “three line”, in the reasonable range of environmental carrying capacity of the development and utilization of water resources. The effective ecological compensation mechanism should be established as soon as possible, so as to achieve long-term comprehensive management of the whole basin, so as to continuously optimize the structure and function of land use.

## Figures and Tables

**Figure 1 ijerph-19-03488-f001:**
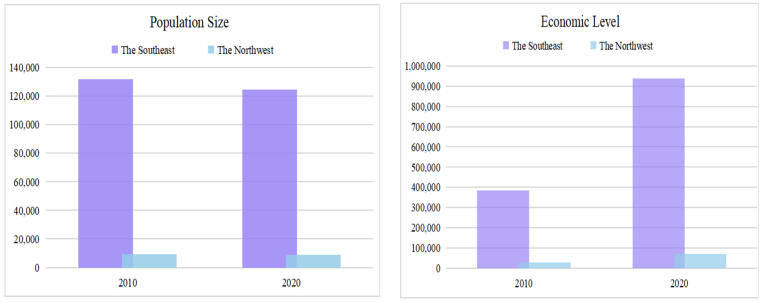
Population size and economic level on both sides of the Hu Line.

**Figure 2 ijerph-19-03488-f002:**
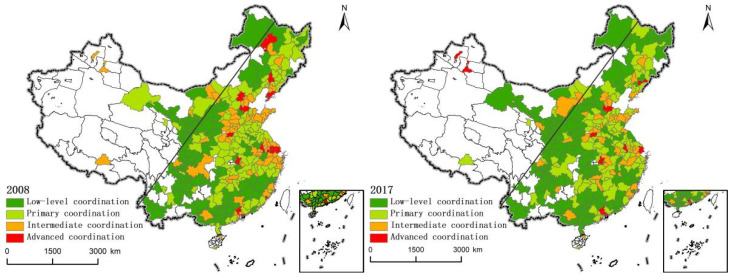
Distribution of “production-living-ecology” functional coordination in China from 2008 to 2017.

**Figure 3 ijerph-19-03488-f003:**
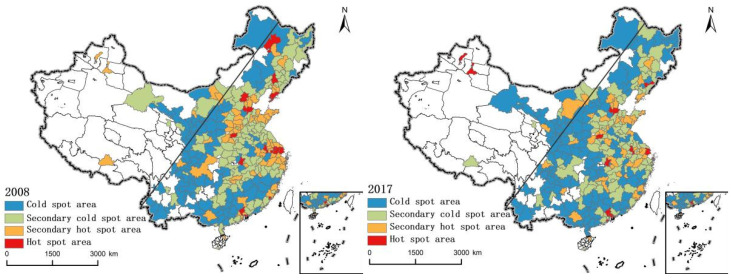
Spatial pattern of “production-living-ecology” functional coordination in China from 2008 to 2017.

**Figure 4 ijerph-19-03488-f004:**
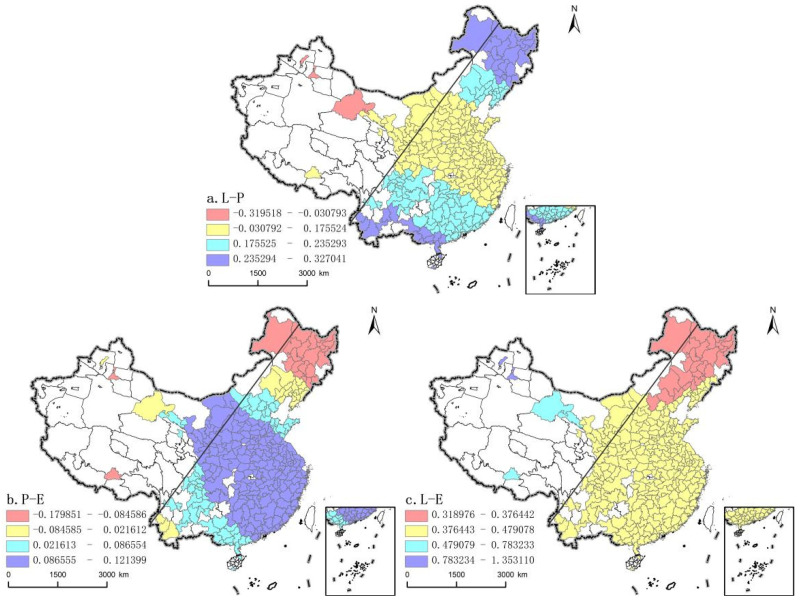
The spatial coefficient distribution of internal driving force (**a**) production-living function, (**b**)production-ecological function, (**c**) living-ecological function.

**Figure 5 ijerph-19-03488-f005:**
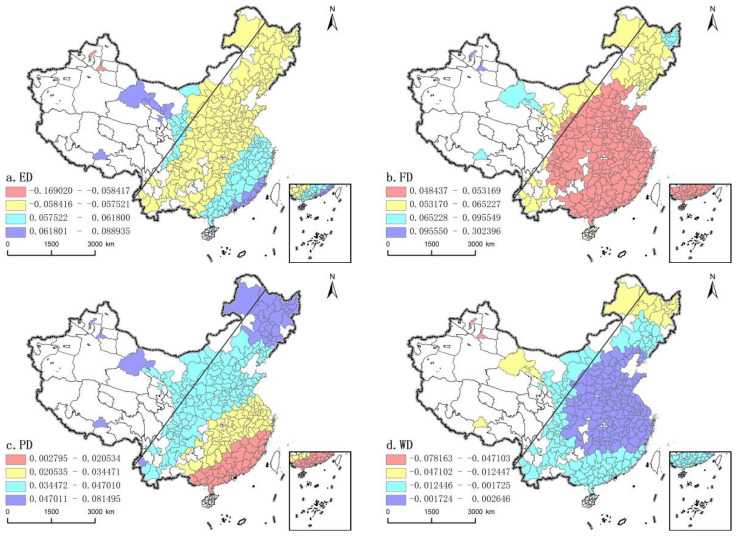
The spatial coefficient distribution of external driving force (**a**) economic density, (**b**) financial density, (**c**) population density, (**d**) water resource density.

**Table 1 ijerph-19-03488-t001:** Functional coordination evaluation index system of “production-living-ecology”.

Criteria Layer	Elements Layer	Basic Indicators
Production function	Agricultural production	Proportion of agricultural land
		Proportion of agricultural output value
		Per unit area yield of grain
	Non-agricultural production	Proportion of construction land
		Average gross industrial output value of land
		Average industrial output value
		Freight volume
	Economic development	Per capita GDP
		Amount of foreign capital used
		Fixed asset investment per land
		The industrial structure
Living function	Living standard	Proportion of residential land area
		Density of road network
	Material life	The employment rate
		Per capita savings balance
	Spiritual life	Proportion of science and education expenditure
		Number of books in public libraries per 10,000 people
		Number of college students per 10,000 persons
Ecological function	Ecological foundation	Green coverage rate of built-up area
		Per capita green garden area
	Ecological carrying	Average industrial wastewater discharge
		Average industrial sulfur dioxide emissions
		Average industrial smoke and dust emission
	Ecological governance	Comprehensive utilization rate general solid waste
		Sewage treatment rate
		Harmless treatment rate of domestic garbage

**Table 2 ijerph-19-03488-t002:** Functional coordination of “production-living-ecology” in China from 2008 to 2017.

	2008	2011	2014	2017
Total	0.1223	0.1732	0.1320	0.1366
The Southeastern	0.1069	0.1701	0.1385	0.1341
The Northwest	0.1207	0.1729	0.1327	0.1364

**Table 3 ijerph-19-03488-t003:** Selection of driving factors for “production-living-ecology” functional coordination.

Driving Factors	Variables	Definition
Internal driving force	Production-Living function	/
	Production-Ecological function	/
	Living-Ecological function	/
External driving force	Population density	population/total area
	Financial density	general budget expenditure/total area of local finance
	Economic density	regional GDP/ total area
	Water resource density	total water resources/total area of the region

**Table 4 ijerph-19-03488-t004:** Calculation results of the GWR model.

	Minimum	Lower Quartile	Mean	Upper Quartile	Maximum
P-L	−0.3195	0.1447	0.1765	0.2098	0.3271
P-E	−0.1798	0.0722	0.0703	0.1111	0.1214
L-E	0.3190	0.3864	0.4041	0.4033	1.3531
ED	−0.1690	0.0563	0.0565	0.0584	0.0889
FD	0.0484	0.0494	0.0537	0.0530	0.3024
PD	0.0028	0.0261	0.0345	0.0433	0.0815
WD	−0.0782	−0.0041	−0.0026	0.0014	0.0026

## Data Availability

The data presented in this study are openly available in National Bureau of Statistics, reference number 978-7-5037-9120-8, 978-7-5037-8770-6, 978-7-5037-8432-3, 978-7-5037-8082-0, 978-7-5037-7706-6, 978-7-5037-7350-1, 978-7-5037-7019-7, 978-7-5037-6754-8.

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
