# Peer review of "Spatiotemporal Evolution and Driving Mechanism of “Production-Living-Ecology” Functions in China: A Case of Both Sides of Hu Line"

_ijerph, 2022, doi:10.3390/ijerph19063488_

Round 1

Reviewer 1 Report

1) The research problem is fairly well described. However, the Introduction section lacks research hypotheses and questions.
2) The theoretical chapter (2.1) or Materials and Methods section (3) lacks a figure that illustrates the course of the Hu Line and its effect on population distribution. The Hu Line should also be indicated in the figures (Figure 3 and 4).
3) Study Area section (3.1) does not quantitatively characterize the set of cities analyzed, nor does it explain on which spatial reference units the analyses were conducted.
4) Section 4.2.2, in its current form, appears to be overly elaborate (pages 18-25). In addition, it contains materials that should be in the Discussion section (the Okun's Law). 
5) The results shown in Figure 5 for several factors (P-E, L-E, ED, FD, WD) seem quite intuitive, especially in interpreting the variation of values with respect to the Hu Line. 
6) The Discussion section (5.2) lacks reference to publications that relate to the issues analyzed in the text.
7) In section 5.2.1 it is worth using bullet points in order to organize the distinctions adopted
8. In section 5.2.2. a classification of industrial types appears. However, the classification is not described before. It is also unclear how the analysed cities are situated in its context.

Reviewer 2 Report

The proposed paper tries to find the correlations and driving mechanism of changing in a large heterogeneous territory as China territory is, using large databases from 288 relevant cities and a decade time series.

The literature review is good, reflecting an intense documentation and also, the used model.

However, there are several shortcomings/weaknesses, as follows:

-the paper has an unusual style which is not appropriate to a journal article, thus the authors need to change their style, providing more synthetic presentation, description, information and not so long description.  The authors are advised, for example, to make their analysis in tabular way, or in a diagrammatic way.  Even the literature review could be more synthetic in a tabular way. In brief, the authors are advised to make their article shorter.

-the developed model and the results must be build-up so that to have a generality, or in other words, to be easy reproduced in other spatial analysis.(here, the authors are kindly asked to pay more attention to the written formula (6) and its explanations/description, because this suggests negligence in writing).

-the mentions on some specific location in China are not relevant for different readers (especial non- Chinese people), thus the authors have to indicate the important locations not by their name but by their special feature in that analysis.

-the conclusion must be more more synthetic, but not again a long and again un-structured discussions.

-even the article title, need to be shorter and more synthetic. 

-some conclusions are trivial (for example, the results of Moran's Index applying doesn't show significant changes, when the time series is rather small, only 10 years, and such changes could be structural changes, which are observable only in larger time scale). The authors need to pay more attention to the conclusions. 
